# The Potential of Dendritic-Cell-Based Vaccines to Modulate Type 3 Innate Lymphoid Cell Populations

**DOI:** 10.3390/ijms24032403

**Published:** 2023-01-26

**Authors:** Lily Chan, Yeganeh Mehrani, Jessica A. Minott, Byram W. Bridle, Khalil Karimi

**Affiliations:** 1Department of Pathobiology, Ontario Veterinary College, University of Guelph, Guelph, ON N1G2W1, Canada; 2Department of Clinical Science, School of Veterinary Medicine, Ferdowsi University of Mashhad, Mashhad 9177948974, Iran; 3ImmunoCeutica Inc., Cambridge, ON N1T 1N6, Canada

**Keywords:** dendritic cell (DC), type 3 innate lymphoid cell (ILC3), innate, communication

## Abstract

Dendritic cell (DC) vaccines are a type of immunotherapy that relies on the communication of DCs with other aspects of the immune system. DCs are potent antigen-presenting cells involved in the activation of innate immune responses and education of adaptive immunity, making them ideal targets for immunotherapies. Innate lymphoid cells (ILCs) are relatively newly identified in the field of immunology and have important roles in health and disease. The studies described here explored the communications between type 3 ILCs (ILC3s) and DCs using a murine model of DC-based vaccination. Local and systemic changes in ILC3 populations following the administration of a DC vaccine were observed, and upon challenge with B16F10 melanoma cells, changes in ILC3 populations in the lungs were observed. The interactions between DCs and ILC3s should be further explored to determine the potential that their communications could have in health, disease, and the development of immunotherapies.

## 1. Introduction

Innate lymphoid cells (ILCs) are a heterogeneous group of leukocytes derived from common lymphoid progenitor cells that have crucial functions in tissue development, repair, and homeostasis. ILCs have been linked to different inflammatory diseases including inflammatory bowel disease, psoriasis, asthma, and various autoimmune diseases [1]. They are typically separated into three main groups, type 1 (ILC1), type 2 (ILC2), and type 3 (ILC3), based on the expression of transcription factors, cytokine profiles, and specific phenotypic markers [2]. Natural killer (NK) cells are sometimes grouped in as a subset of ILC1s due to their similarities; however, NK cells have been shown to have distinct features from ILC1s such as transcription factor expressions and cytolytic activity that have distinguished them as their own subtype of ILCs [3]. However, many still consider NK cells as a subset of ILC1s [4]. Due to their similar cytokine profiles, transcription factors, and immunological functions, ILCs (NK cells, ILC1s, ILC2s, and ILC3s) are considered to be innate counterparts to components of the adaptive immune system mirroring cytotoxic T lymphocytes, Th1, Th2, and Th17 responses, respectively [4]. However, unlike T cells, they lack antigen-specific receptors [5].

DCs are innate leukocytes that are critical in the induction of immune responses and tolerance. They have pattern recognition receptors that recognize pathogen- and/or damage-associated molecular patterns and are the most efficient antigen-presenting cells [6]. Through antigen presentation and cytokine secretion, DCs can dictate naïve T cells’ fate and induce pro-inflammatory responses or immunosuppression. T cells can be educated to recognize an antigen as dangerous and attack it or as non-dangerous and take on a tolerogenic phenotype [6]. DCs are also involved in the activation of innate immunity as well as the education of adaptive immunity. Through direct contact or the production of different cytokines and soluble factors, DCs can activate other innate leukocytes and promote innate immune responses [7].

The power of DCs has been utilized in immunotherapies such as DC vaccines. DC vaccines are prepared by isolating DCs from a patient or isolating DC precursors and deriving DCs. These DCs are manipulated ex vivo, which involves the maturation of the DCs in an immunogenic manner. The DCs are also loaded with specific antigens, which will be what the vaccine will aim to develop immunogenic responses directed towards [8]. DC vaccines are an immunotherapeutic platform that takes advantage of the ability of DCs to educate the adaptive immune system to induce antigen-specific responses, particularly antigen-specific cytotoxic T cells. Therefore, the majority of DC vaccine research has focused on the capacity of DCs to educate the adaptive immune system. However, in recent years, the relationship between DCs and NK cells has been shown to have a powerful influence in anti-tumor immunity which is crucial for the efficacy of DC vaccination [9,10,11]. This demonstrates the important need to explore other relationships between DCs and different innate leukocytes and the influence these communications can have on the efficacy of DC vaccines. Since NK cells are a part of the ILC family and due to the close resemblance of ILCs to T cells [5], the objective of the studies presented here was to investigate the potential relationship between DCs and another member of the ILC family, ILC3s. We used a monocyte-derived DC (moDC) vaccine platform where DCs were generated ex vivo from murine bone marrow precursor cells. MoDC vaccines are the most common form of DC vaccines as, historically, they were the easiest to produce [12].

ILC3s have important functions in intestinal homeostasis, immunity against extracellular bacteria, and the development of lymphoid tissues [5]. ILC3s can be subdivided into groups that vary in their expression of surface markers and cytokine production. However, all ILC3s express RORγt [13]. One subgroup of ILC3s is ILC3-lymphoid tissue inducer cells which are important in organogenesis [14]. ILC3s can also be subdivided based on natural cytotoxicity receptor (NCR) expression, which in mice is NKp46 expression [15,16] and in humans, it is NKp44 [17,18]. Therefore, there are NCR^+^ and NCR^−^ ILC3s that have differences in their functional capacities and phenotypic characteristics. For instance, it has been suggested that NCR^+^ ILC3s produce IL-22 but not IL-17, whereas NCR^−^ ILC3s can produce both cytokines [19]. However, in a recent study by Fiancette et al., NCR^+^ ILC3s were shown to be able to produce a low amount of IL-17 [20]. Rankin et al. demonstrated in mice that NCR^+^ ILC3s require the transcription factor T-bet. Therefore, NCR^+/−^ ILC3s can be differentiated based on NKp46 and T-bet expression. In the studies described here, murine ILC3s were defined by lineage^−^CD45^+^DX5^−^RORγt^+^, with NCR^+^ ILC3s also being T-bet^+^NKp46^+^ and NCR^−^ILC3s being T-bet^−^NKp46^−^ (Table 1 and Appendix A Figure A1, Figure A2 and Figure A3).

This research’s objective was to help elucidate if there are communications between DC vaccines and ILC3s and the potential their interactions could have in DC immunotherapies. ILC3s have been demonstrated to have roles in various diseases and cancers and therefore have potential in different immunotherapy contexts. ILC3s are relatively new to the scene of immunology and as a result, their influences in disease and treatments have been explored limitedly. ILC3s respond to external stimuli, which will dictate their activities and whether they promote the progression or suppression of tumors [21,22]. Similar to Th17 cells, ILC3s have the transcription factor RORγt and produce Th17 response cytokines such as IL-17 and IL-22. IL-22 is important for controlling bacterial infections in the gut [16]. However, in a model of bacteria-induced colon cancer, it was demonstrated that IL-17 and IL-22 from ILCs in the colon contributed to the development of the colon cancer [23]. Furthermore, ILC3s have been correlated with negative outcomes in breast cancers [24]. Conversely, an association between NCR^+^ ILC3s and tertiary lymphoid structures (TLS) and better clinical outcomes in non-small cell lung cancers have been observed [18]. Additionally, a recent study investigated ILC3s in colorectal cancer. The study observed an increase in ILC1s and a decrease in ILC3s in patient samples of resected colorectal tumors compared to matched non-malignant adjacent tissues. RNA sequencing and transcriptional profiling revealed ILC3s to have increased plasticity and different functional capacities within colorectal tumors compared to non-malignant tissues. Interestingly, RNA sequencing also suggested that the tumor-infiltrating ILC3s had upregulated plasticity to shift to an ILC1 phenotype. The study observed that ILC3 interactions with T cells supported type I immunity, and in colorectal cancers, these interactions are limited, which hinders anti-tumor responses. This supports an important function for ILC3s in anti-tumor immunity. Overall, the paper demonstrated the significant roles that ILC3s have in the regulation of immunologic homeostasis and colorectal cancer [25]. Therefore, ILC3s are considered to have dual roles in health and disease. They can promote cancer progression [23,24,26] but can also contribute to anti-tumor responses [18,25,27,28]. Their phenotype and contributions to cancer appear to be contextual, which indicates a potential to manipulate ILC3s via immunotherapy to obtain a phenotype that assists with anti-cancer responses rather than supporting tumor progression. Thus, DC-based vaccines represent a possible method to influence the tumor microenvironment through manipulation of ILC phenotypes and their cytokine production, which would allow for tailoring of the immune response depending on the circumstances and could be beneficial in cancer research.

Since there has been limited research into the communications between DCs and ILC3s, especially in the context of DC vaccination, the studies described here investigated the changes in ILC3 populations following administration of a DC-based vaccine to mice. In this paper, communication between a DC-based vaccine and ILC3s was observed, which was shown through the sustained influence of the vaccine on ILC3 responses for at least 10 days post-immunization.

## 2. Results

### 2.1. The Numbers of Both NCR^+^ and NCR^−^ ILC3s Increased in the Local Draining Lymph Node after Administration of DC Vaccines

A DC vaccine was prepared by differentiating DCs from murine-bone-marrow-derived cells ex vivo. These DCs were stimulated to promote their maturation so they would acquire an immunogenic phenotype. The DCs were then loaded with a peptide. Since this study focuses on the nature of communication between two innate leukocytes, DCs and ILC3s, the peptide selected for production of the vaccine was intentionally irrelevant to our biological model to avoid antigen-specific T cells becoming a confounding variable. Therefore, the DC vaccine was loaded with a chicken-ovalbumin-derived peptide.

To begin investigating if DC vaccination influences ILC3 populations, we first evaluated if there were differences in cellularity proximal to the vaccination site. There were changes in the local draining lymph node following DC vaccination, which included an in increase in the number of ILC3s (Figure 1). The kinetics of the increase in ILC3s showed that ILC3s gradually increased following DC vaccination (Figure 1b,c). The number of NCR^+^ and NCR^−^ ILC3s in the local draining lymph node was assessed and both subpopulations increased following the administration of the DC vaccine. Although both NCR^+^ and NCR^−^ ILC3 populations peaked at three days post-DC vaccination, NCR^−^ ILC3s returned to homeostatic numbers seven days post-immunization while the number of NCR^+^ ILC3s remained significantly higher compared to sham-treated control mice.

### 2.2. DC Vaccines Increased ILC3 Cytokine Production in the Spleen without Changing the Total Number of Splenic ILC3s

Next, systemic changes in ILC3 populations were examined by evaluating the spleen, which is the largest secondary lymphoid organ [29], one week following DC vaccination, which is our laboratory’s standard time point for evaluating changes in leukocytes systemically. There was no significant difference observed in the total number of NCR^+^ or NCR^−^ ILC3s in the spleen (Figure 2).

Since ILC3 cytokine profiles can be influenced by environmental factors, ILC3 production of IL-22 and IL-17 in the spleen was measured one week following DC vaccination. There was an increase in ILC3s producing IL-17 or IL-22 or both cytokines in mice treated with DC vaccines compared to the control mice (Figure 3).

### 2.3. ILC3 Subpopulations Following DC Immunization and Challenge with B16F10 Melanoma Cells

ILC3s can play dual roles in cancer progression by contributing to both pro- and anti-tumorigenic responses [22]. Therefore, the influence of DC vaccination on ILC3 responses in a cancer context was investigated. Mice were treated with DC vaccines and one week later, they were intravenously challenged with B16F10 melanoma cells, which resulted in seeding of the lungs. As mentioned, since only components of innate immunity were being assessed, the antigen-specific education by the vaccine was not to be assessed in our experiments and, therefore, the melanoma model used did not express ovalbumin, rendering the peptide loaded onto the DCs irrelevant. The kinetics experiment (Figure 1) demonstrated that local responses of ILC3s could be observed within three days. Therefore, three days post-challenge, the number of ILC3 subpopulations and their ability to produce cytokines were examined in the spleens and lungs. Similar to what was observed in the spleens of naïve mice, there was no significant difference in the total number of splenic NCR^+^ and NCR^−^ ILC3s between the control group and the DC-immunized group (Figure 4). However, unlike in the naïve model, in the spleen, there were no longer changes in the production of IL-17 and/or IL-22 (Figure 5).

Intravenous injection of B16F10 cells via the tail vein is a common mouse model of synthetic melanoma metastases to the lungs [30]. Therefore, the lungs were also examined to determine the number of ILC3 subpopulations and assess their production of cytokines. There was a decrease in the number of NCR^−^ ILC3s with a concomitant increase in the number of NCR^+^ ILC3s (Figure 6). However, there was no observed effect of the DC vaccination on ILC3-mediated cytokine production in the lungs (Figure 7).

## 3. Discussion

The studies described here sought to investigate potential communications between ILC3s and DCs in the context of a DC vaccine. There were increases in the number of both NCR^+^ and NCR^−^ ILC3 subpopulations in draining lymph nodes (Figure 1) following DC vaccination, which was suggestive of local communications occurring between the DC vaccine and ILC3 subpopulations. However, the communication between DCs and NCR^+^ ILC3 had a longer-lasting influence on the total number of NCR^+^ ILC3s than that observed between DCs and NCR^−^ ILC3s. This indicated that DCs may have unique interactions with ILC3s depending on their expression of NCRs. There was no change observed in the number of ILC3 subpopulations in spleens seven days following DC vaccination (Figure 2) or ten days following DC vaccination and three days post-B16F10 melanoma cell challenge (Figure 4). Our methods to assay ILC3s for their IL-17 and IL-22 production have a limitation because they show both the translational and transcriptional responses due to in vitro restimulation of cells. However, we added an experimental control in all of the experiments, which would be the no-stimulation treatment in vitro to detect what ILC3s produce in response to the signal received in vivo. As illustrated in Figure 3d, Figure 5d and Figure 7d, the cells that received no in vitro stimulation produced IL-17 and IL-22. In addition, although splenic ILC3 cell numbers did not change (Figure 2), cytokine production by the ILC3 subpopulations was different seven days following DC vaccination (Figure 3). This suggested the DC vaccination could influence the functionality of ILC3 responses without affecting them numerically. Being components of the innate immune system, trafficking of ILC3s in and out of tissues could have occurred in the span of a few days, therefore preventing detection of treatment-modulated numerical differences seven to ten days following DC vaccination. Evidence for this was observed in the DC-vaccine-draining lymph node where the number of ILC3s increased over the course of three days and then began decreasing (Figure 1). Thus, there may have been numerical changes in ILC3 subpopulations in the spleen that were missed because they had returned to homeostatic numbers by the seventh day post-DC immunization. Nonetheless, the studies were focused on longer lasting responses as opposed to short-term transient responses and, therefore, did not investigate this avenue further. Interestingly, although the number of splenic ILC3 subpopulations was not different from the control groups, their cytokine profiles were altered (Figure 3). Production of IL-22 and IL-17 were increased in splenic ILC3s, thus demonstrating that DC vaccination had a systemic effect on ILC3s. In summary, DC vaccination did not impact the quantity of splenic ILC3s but did influence their functionality.

Although there was a sustained effect on ILC3-derived cytokine production seven days post-DC immunization in the spleens of tumor-free mice, this influence on ILC3s in the spleen was undetectable three days following intravenous challenge with B16F10 melanoma cells (Figure 5). The DC vaccine appeared to prime ILC3 populations in the tumor-free model that was suppressed upon tumor challenge. Since ILCs are greatly influenced by their environment, this could suggest that the communications between the DCs in the vaccine and ILC3s was limited to the tumor-free model. Conversely, this could also be due to the mobilization of primed ILC3s to other locations in the body where the B16F10s may have congregated and established a microenvironment that induced leukocyte recruitment.

There was a shift in ILC3 subpopulations observed in the lungs ten days post-DC vaccination and three days post-B16F10 cell challenge compared to the non-DC-vaccinated control mice that only received administration of B16F10 cells. Specifically, there was an increase in the number of NCR^+^ ILC3s and a decrease in NCR^−^ ILC3s in the lungs (Figure 6). T-bet is a transcription factor that is associated with Th1 responses [31]. Therefore, the increase in NCR^+^ILC3s, which express T-bet, indicated a potential promotion of type 1 immunity. This could be a beneficial ILC3 response in a cancer context where type 1 immune responses are usually desired.

It has been shown that NCR^+^ ILC3s can derive from a subset of NCR^−^ ILC3s [16]. Therefore, the increase in NCR^+^ ILC3s and the decrease in NCR^−^ ILC3s could be attributed to a stimulation that induced a switch in NCR^−^ ILC3s to become NCR^+^ ILC3s. If the ILC3s were undergoing phenotype switching from NCR^−^ to NCR^+^ they may not have been able to produce cytokines during or shortly after switching and while adjusting to the surroundings and their new phenotype. Therefore, their cytokine production may have been halted or delayed while phenotype switching was occurring, which could explain why there were no changes in ILC3-mediated IL-17 and/or IL-22 production in the lungs (Figure 7). Though, it is also possible that DC vaccination simply did not influence cytokine production by ILC3s in the lungs at this timepoint.

## 4. Materials and Methods

Ethics Approval

All of the murine studies were performed following the animal utilization protocol #3807 under the supervision of the animal care staff at the University of Guelph.

Mice

Female C57BL6 mice were received from Charles River Laboratories aged five to eight weeks. The mice were housed in a controlled environment at the University of Guelph’s animal isolation unit and given one week to acclimate prior to commencement of the experiments. The mice were fed and given water ad libitum.

Dendritic Cell Cultures

The femurs and tibias of female C57BL6 mice were harvested and the ends of the bones were cut off. Using a syringe, the bone marrow of the tibias and femurs were flushed out with PBS into a Petri dish. The bone marrow was resuspended into a single-cell suspension and was counted using a hemocytometer. The cells were then resuspended in media (RPMI [HyClone Cat# SH30027.01] with 2-mercaptoethanol [Gibco Ref# 21985-023], 1% penicillin/streptomycin [HyClone Cat# SV30010], and 10% fetal bovine serum [VWR Cat#97068-085]) to a concentration of 1.25 × 10^6^ cells per mL, supplemented with 20 ng/mL granulocyte–macrophage colony-stimulating factor (GM-CSF) (Biolegend Cat#576308) and aliquoted into 25 cm^2^ culture flasks, 5 mL per flask. The cultures were put into humidified incubators at 37 °C with 5% CO_2_ and allowed to grow for 7 days. On day two of the culture, 5 mL of fresh media with 20 ng/mL GM-CSF was added. On day five, 5 mL of each culture was centrifuged, and the supernatant was removed. The cells were resuspended in 5 mL of fresh media with 20 ng/mL GM-CSF and re-added to the culture flasks. The cultures were harvested on day 7 and prepared for vaccination.

Dendritic Cell Vaccination Preparation

The dendritic cell (DC) cultures were transferred to a 50 mL conical tube and counted using a hemocytometer. The cells were stimulated with 100 ng/mL lipopolysaccharide (LPS) from Escherichia coli O55:B5 (Sigma Cat#L2880) and 1 μg/mL of chicken ovalbumin (OVA)_257–264_(SIIN) (PepScan Systems, Lelystad, Netherlands) for 1 h in an incubator at 37 °C with 5% CO_2_. The cells were then washed with PBS three times and resuspended to a concentration of 5 × 10^5^ cells per 30 µL. The vaccines were administered at doses of 5 × 10^5^ cells per 30 μL PBS.

Tissue Processing

The lymph nodes and spleens were harvested and placed into Petri dishes with 2 mL Hanks buffered saline solution (HBSS). They were pressed into single-cell suspensions using the back of a 3 mL syringe stopper. The single cell suspensions were filtered into a 50 mL conical tube using a 70 μm-pore-size cell strainer. The lymph nodes were counted using a hemocytometer. The spleens were centrifuged, the supernatant was removed, and the cells were resuspended in ACK lysis buffer (8.29 g NH_4_Cl [0.15 M], 1 g KHCO_3_ [10.0 mM], 37.2 mg Na_2_EDTA [0.1 mM] in 1 mL of Milli Q water) and left to sit for five minutes to lyse the erythrocytes. The cells were washed with HBSS twice before being resuspended in media and counted using a hemocytometer.

The lungs were harvested and weighed before being placed in a gentleMACS^TM^ tube containing 1 mg/mL collagenase IV (Gibco Ref#17104-019) and 5 μg/mL DNase I (Roche Ref#11284932001) in HBSS. Using the gentleMACS^TM^ dissociator, the samples were run through lung protocol A and then incubated for 20 min in an incubator at 37 °C with 5% CO_2_. The samples were then run through lung protocol B on the gentleMACS^TM^ dissociator. Twice the sample volume of HBSS was added to the samples to neutralize the enzyme reaction. The samples were filtered into a 50 mL conical tube through a 70 μm-pore-size cell strainer. The cells were then washed twice with HBSS and resuspended in ACK lysis buffer. After five minutes in ACK lysis buffer, the cells were washed twice with HBSS and resuspended in media.

Cytokine Response Assay

The single-cell suspensions from the tissue samples were seeded into 96-well plates in duplicates, where one well was treated as the no-stimulation control and the other well received a stimulant treatment. The unstimulated wells were only given media and the stimulant-treated wells received 10 ng/mL phorbol myristate acetate (PMA) and 1,500 ng/mL ionomycin in media. The cells were placed in an incubator at 37 °C with 5% CO_2_ for one hour. Then, Brefeldin A (GolgiPlug, Biolegend Cat#420601) (×100 dilution) was added to all of the sample wells, and the plate was placed back into the incubator for an additional four hours. The cells were then washed twice with PBS and stained for analysis by flow cytometry.

B16F10 melanoma cells

B16F10 cells were thawed from liquid nitrogen and counted using a hemocytometer. The cells were then resuspended in media (Dulbecco’s high glucose modified Eagles medium [HyClone Cat#SH3002201] with 1% penicillin/streptomycin [HyClone Cat# SV30010] and 10% bovine calf serum [VWR Cat#10158-358]) to a concentration of 1 × 10^5^ cells per mL. The B16F10 cells were then aliquoted into culture flasks and placed into an incubator at 37 °C with 5% CO_2_. To prepare the B16F10 cells for administration, the cells were transferred from the culture flasks to a 50 mL conical tube, washed with PBS, and then resuspended in PBS. The cells were counted using a hemocytometer and resuspended in PBS to give doses of 3 × 10^5^ cells per 200 μL.

Intracellular Cytokine Antibody Staining

The cells were seeded into 96-well plates and resuspended in Fc block (anti-CD16/CD32, BioLegend Cat#101320) and incubated at 4 °C for 15 min. The cells were washed with phosphate-buffered saline containing 0.5% bovine serum albumin (FACS buffer) twice, then re-suspended in a master mix of cell surface-staining antibodies (CD45, BioLegend Cat#103132; NKp46, BioLegend Cat#137617; DX5, BioLegend Cat#108919; CD127, BioLegend Cat#135007; Lineage Cocktail, BioLegend Cat#133311) and incubated at 4℃ for 20 min. The cells were washed with phosphate-buffered saline twice and resuspended in Zombie Aqua Fixable viability dye (FVD) (BioLegend Cat#423101) and incubated at 4 °C for 30 min. Phosphate-buffered saline was used to wash the cells twice before resuspension in fixation buffer (BioLegend Cat#420801). The cells were incubated at 4 °C for 20 min. After fixation, the cells were washed twice with permeabilization buffer (BioLegend Cat#421002). The cells were resuspended in a master mix of intracellular-staining antibodies (IL-22, BioLegend Cat#516404; IL-17A, eBioscience Cat# 17-7177-81) and incubated at 4 °C for 20 min. The cells were washed twice with permeabilization buffer, then resuspended in FACs buffer, and then processed with a BD FACSCantoTM II flow cytometer and analyzed using BD FACSDivaTM software.

Transcription Factor Antibody Staining

The cells were seeded into 96-well plates and resuspended in a FC block (anti-CD16/CD32 BioLegend Cat# 101320) and incubated at 4 °C for 15 min. The cells were washed twice with FACs buffer (0.5% in bovine serum albumin [HyClone Cat# SH30574.02] in PBS), then re-suspended in a master mix of cell surface-staining antibodies (CD45, BioLegend Cat#103132; NKp46, BioLegend Cat#137617; DX5, BioLegend Cat# 108919; CD127, BioLegend Cat#135007; Lineage Cocktail, BioLegend Cat#133311), and incubated at 4 °C for 20 min. The cells were then washed with phosphate-buffered saline twice and resuspended in FVD (BioLegend Cat#423101) and incubated at 4 °C for 20 min. The cells were washed with phosphate-buffered saline twice and resuspended in mouse FoxP3 fixation buffer (BD Pharmingen BD Sciences Cat#51-9006124) and incubated at 4 °C for 30 min. After fixation, the cells were washed with mouse FoxP3 permeabilization buffer (BD Pharmingen BD Sciences Cat# 51-9006125) and resuspended in FoxP3 permeabilization buffer and incubated at 37 °C for 30 min. The cells were resuspended in transcription factor antibodies (ROR gamma(t), eBioscience Cat#17-6988-82; T-bet, eBioscience Cat# 12-5825-82) and incubated at 4 °C for 20 min. The cells were washed with permeabilization buffer twice, then resuspended in FACs buffer, and were analyzed using a BD FACSCantoTM II flow cytometer and BD FACSDivaTM software.

Statistical analysis

Splenic and pulmonary populations of NCR^+^ and NCR^−^ ILC3s were analyzed using unpaired t-tests. The kinetic experiment population analysis was compared using ordinary one-way analysis of variance (ANOVA) and Tukey’s multiple comparison test. The cytokine analysis in the spleens and lungs was examined using two-way ANOVA and Šídák’s multiple comparisons test. Means for the treatment groups were defined as significantly different with a *p*-value < 0.05.

Gating strategy

First, using forward scatter area (FSC-A) and side scatter area (SSC-A), the lymphocytes were gated on. Then, doublet cells were excluded using FSC-A and FSC-height (FSC-H). Live cells were gated on by taking the cells negative for FVD. Then, CD127^+^ and lineage cocktail cells were taken to narrow down on the ILCs. The CD45^+^ cells were then gated on. To ensure that all of the NK cells were excluded, DX5^−^ was then gated on. For transcription factor identification of ILC3s, RORγt^+^ cells were then gated on and from these cells, NCR^+^ ILC3s were T-bet^+^ and NKp46^+^ and NCR^−^ ILC3s were T-bet^−^ and NKp46. For cytokine identification, following gating on lymphocytes, single cells, live cells, and CD127^+^lineage^−^CD45^+^, IL-17, and IL-22 were then gated on as being produced by ILC3s since ILC3s would be the only cells in this final gate that could be producing IL-22 and IL-17.

## 5. Conclusions

Although the increased ILC3-derived IL-17 and IL-22 production in the spleen observed in the tumor-free model was lost in the tumor challenge model, there was an opposite shift in NCR^+^ and NCR^−^ ILC3 populations in the lungs. This suggests that DC vaccination could have influenced ILC3s in the tumor challenge model as well as in the tumor-free mice. However, the effect of this relationship between the DCs in the vaccine and ILC3s on anti-tumor responses remains unknown and is a future research direction. Thus, the studies described here demonstrate that there are communications occurring between DC-based vaccines and ILC3 subpopulations, both in tumor-free and tumor-bearing mice. These communications and their subsequent impacts on immune responses appear to be complex. These observations contribute to the body of literature seeking to decipher the relationship between DC vaccines and ILC3s in an effort to develop enhanced DC vaccination strategies. We recommend further studies investigating the potential to design vaccination strategies that can modulate these communications to optimize ILC3 responses to assist in anti-tumor responses and limit support of pro-tumor responses.

## Figures and Tables

**Figure 1 ijms-24-02403-f001:**
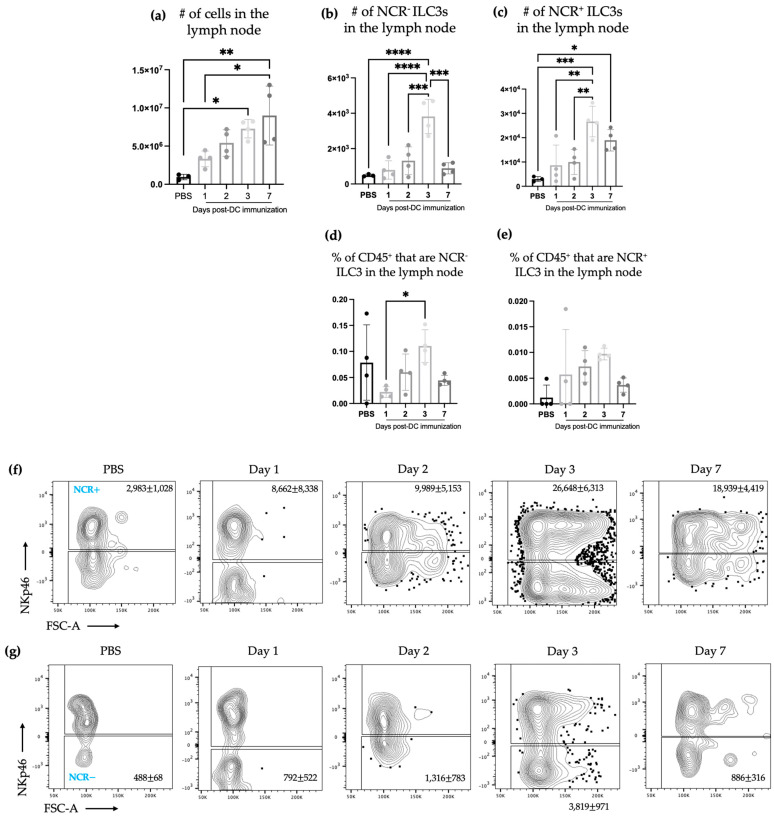
The number of natural cytotoxicity receptor (NCR)^+^ and NCR^−^ type 3 innate lymphoid cells (ILC3s) increased in the local draining lymph node after DC immunization. Female C57BL/6 mice were inoculated with DC vaccines via hind footpad injections. Popliteal lymph nodes were examined for ILC3 populations. (**a**) The total number of cells in the lymph node was determined. Accumulation of (**b**) NCR^−^ ILC3s and (**c**) NCR^+^ ILC3s in the lymph node and the percentage of CD45^+^ cells in the lymph node that were (**d**) NCR^−^ ILC3s and (**e**) NCR^+^ ILC3s. Each bar represents data from four popliteal lymph nodes. A Student’s *t*-test was used at each time point to determine significant differences between the control mice inoculated with phosphate-buffered saline (PBS) and the mice inoculated with the DC vaccine (*p*-values * < 0.05, ** < 0.005, *** < 0.0005, and **** < 0.0001). Representative dot plots showing the average number of (**f**) NCR^−^ ILC3s and (**g**) NCR^+^ ILC3s in the draining popliteal lymph node. The graphs display the mean with the standard deviation.

**Figure 2 ijms-24-02403-f002:**
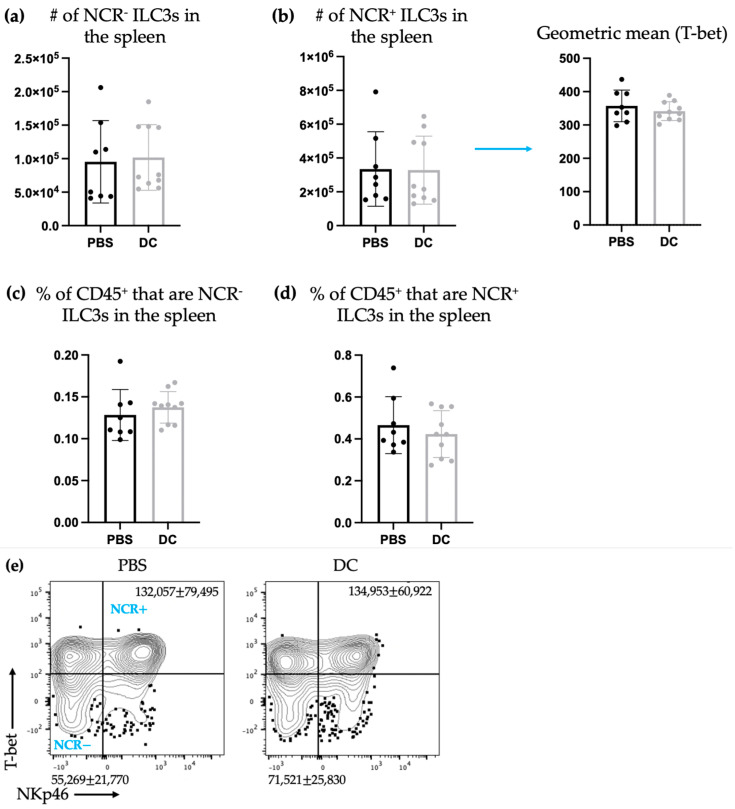
There were no changes in the number of splenic ILC3s after DC immunization. Female C57BL/6 mice (*n* = 8 [PBS] or 10 [DC]) were inoculated with DC vaccines via hind footpad injections. Spleens were harvested one week following inoculation and examined for ILC3 populations. Splenic total number of (**a**) NCR^−^ ILC3s and (**b**) NCR^+^ ILC3s (and the geometric mean fluorescent intensity of T-bet for NCR^+^ ILC3s) and the percentage of splenic CD45^+^ cells that were (**c**) NCR^−^ ILC3s and (**d**) NCR^+^ ILC3s were monitored and quantified by flow cytometry analysis. A Student’s t-test was used to determine significance between the populations in the control mice inoculated with phosphate-buffered saline (PBS) and the DC-inoculated mice. The means were not significantly different. Representative dot plots showing the average number of splenic (**e**) NCR^−^ ILC3s and NCR^+^ ILC3s. The standard deviation and mean are represented by the graph bars and error bars.

**Figure 3 ijms-24-02403-f003:**
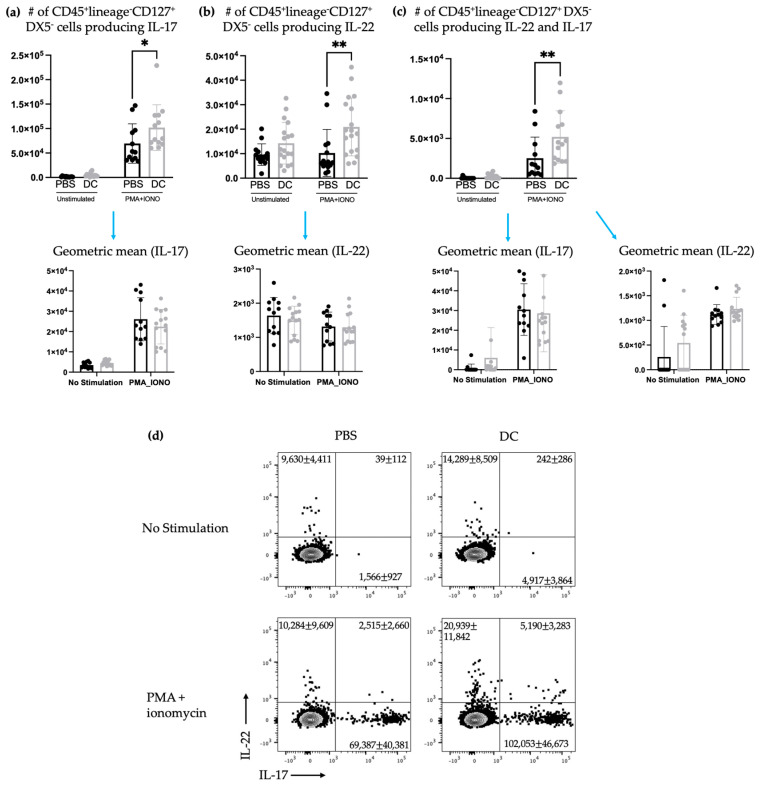
There was an increase in splenic ILC3s producing IL-17 and IL-22 following DC immunization. Female C57BL/6 mice (*n* = 12–18) were inoculated with DC vaccines via hind footpad injection. The control mice were treated with phosphate-buffered saline (PBS). One week after immunization, spleens were harvested and examined for IL-17- and IL-22-producing ILCs. The number of lineage^−^CD127^+^DX5^−^ cells producing (**a**) IL-17, (**b**) IL-22, and (**c**) both IL-17 and IL-22 and their corresponding geometric mean fluorescent intensities was determined using intracellular cytokine staining and flow cytometry. Data were analyzed using a two-way ANOVA test (*p*-values * < 0.05, ** < 0.005). (**d**) Representative dot plots show the average number of splenic CD45^+^lineage^−^CD127^+^DX5^−^ cells producing IL-17 and/or IL-22. The graphs display the mean with the standard deviation.

**Figure 4 ijms-24-02403-f004:**
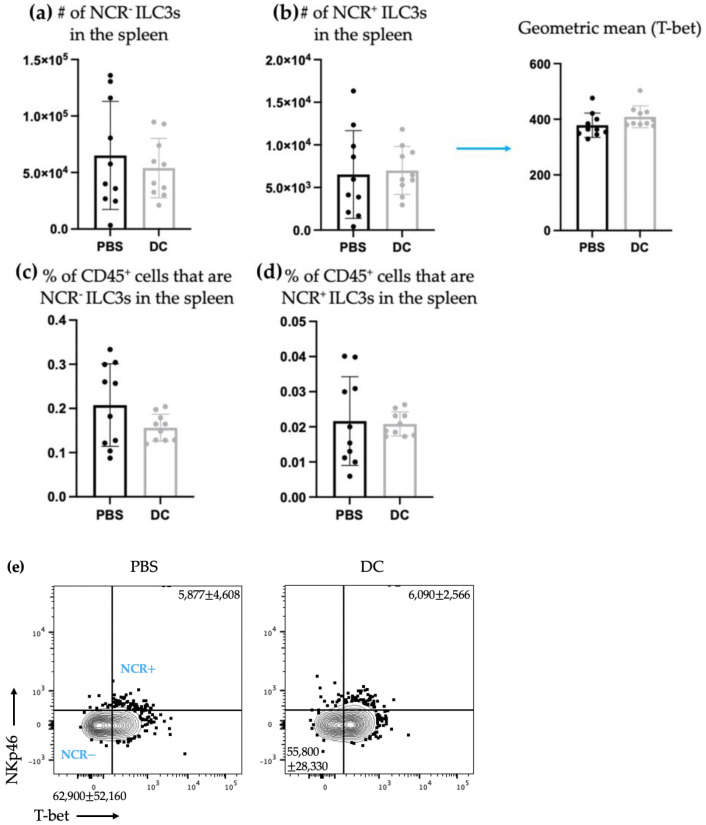
There was no change in the total number of splenic ILC3s after DC vaccination and challenge with B16F10 melanoma cells. Female C57BL/6 mice (*n* = 10) were inoculated with DC vaccines via hind footpad injection and one week later, the mice were administered with 3 × 10^5^ B16F10 cells via tail vein injection. Spleens were harvested and examined for the accumulation of ILC3 populations three days after B16F10 administration. The number of splenic (**a**) NCR^−^ ILC3s and (**b**) NCR^+^ ILC3s (and the geometric mean fluorescent intensity of T-bet for NCR^+^ ILC3s) and the percentage of splenic CD45^+^ cells that were (**c**) NCR^−^ ILC3s and (**d**) NCR^+^ ILC3s were monitored and quantified by flow cytometry. A Student’s *t*-test was used to determine significance between the populations in the control mice treated with phosphate-buffered saline (PBS) and the DC-inoculated mice. The means were not significantly different. Representative dot plots show the average number of splenic (**e**) NCR^−^ ILC3s and NCR^+^ ILC3s. The mean and standard deviation are represented with the bar graphs and error bars.

**Figure 5 ijms-24-02403-f005:**
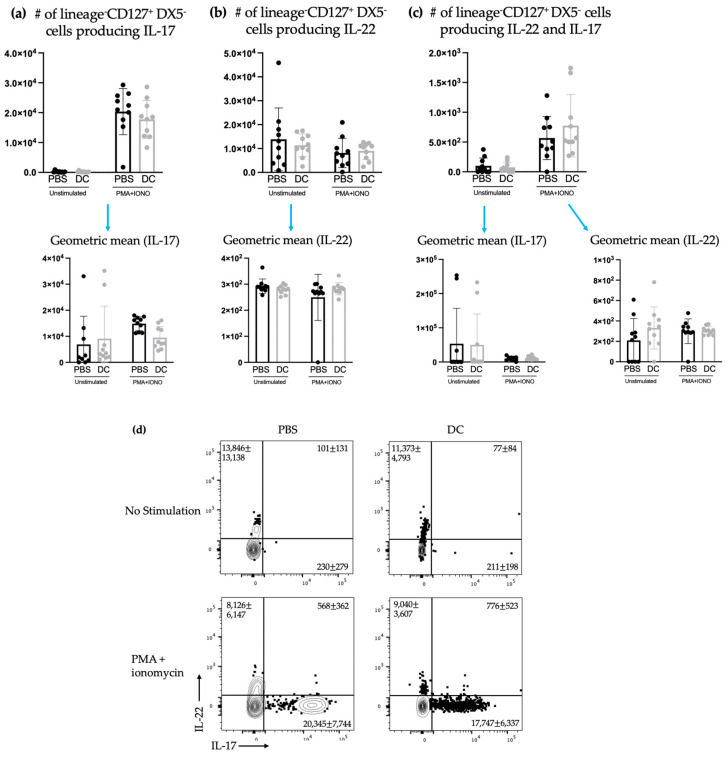
There was no change in the number of splenic IL-17 and/or IL-22-producing ILCs after DC vaccination and challenge with B16F10 melanoma cells. Female C57BL/6 mice (*n* = 10) were inoculated with DC vaccines via hind footpad injection and one week later, 3 × 10^5^ B16F10 cells were administered intravenously. Three days later, spleens were harvested and examined for ILC3 cytokine productions. The number of lineage^−^CD127^+^DX5^−^ cells producing (**a**) IL-17, (**b**) IL-22, and (**c**) both IL-17 and IL-22 and their corresponding geometric mean fluorescent intensities was determined using intracellular cytokine staining and flow cytometry. Data were analyzed using a two-way ANOVA test and no significant difference between the DC-vaccinated mice and the phosphate-buffered saline (PBS)-treated controls was detected. (**d**) Representative dot plots show the average number of splenic CD45^+^lineage^−^CD127^+^DX5^−^ cells producing IL-17 and/or IL-22. The bar graphs and error bars display the mean and standard deviation.

**Figure 6 ijms-24-02403-f006:**
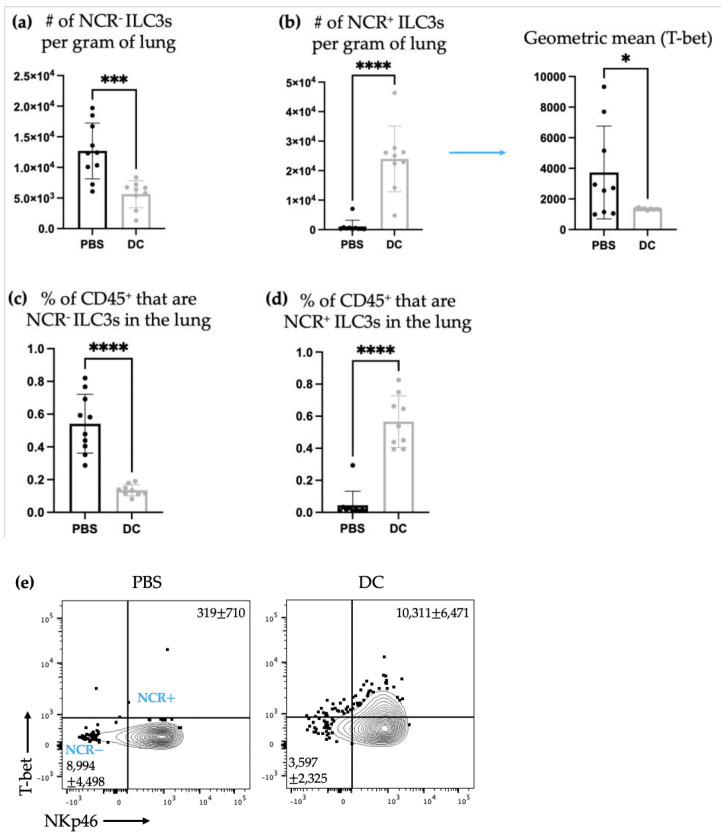
Following DC immunization and challenge with B16F10 cells, there were numerical changes in ILC3 subpopulations in the lungs. Female C57BL/6 mice (*n* = 10) received DC vaccines via hind footpad injection and one week later, 3 × 10^5^ B16F10 cells were administered intravenously. Three days post-challenge, the lungs were examined using flow cytometry for the number of (**a**) NCR^−^ ILC3s and (**b**) NCR^+^ ILC3s (and the geometric mean fluorescent intensity of T-bet for NCR^+^ ILC3s) and the percentage of CD45^+^ cells that were (**c**) NCR^−^ ILC3s and (**d**) NCR^+^ ILC3s. A Student’s *t*-test was used to determine significance between the subpopulations in the control mice treated with phosphate-buffered saline (PBS) and the DC-inoculated mice (*p*-values * <0.05, *** <0.0005, and **** < 0.0001). Representative dot plots showing the average number of pulmonary (**e**) NCR^−^ ILC3s and NCR^+^ ILC3s. The graphs display the mean with the standard deviation.

**Figure 7 ijms-24-02403-f007:**
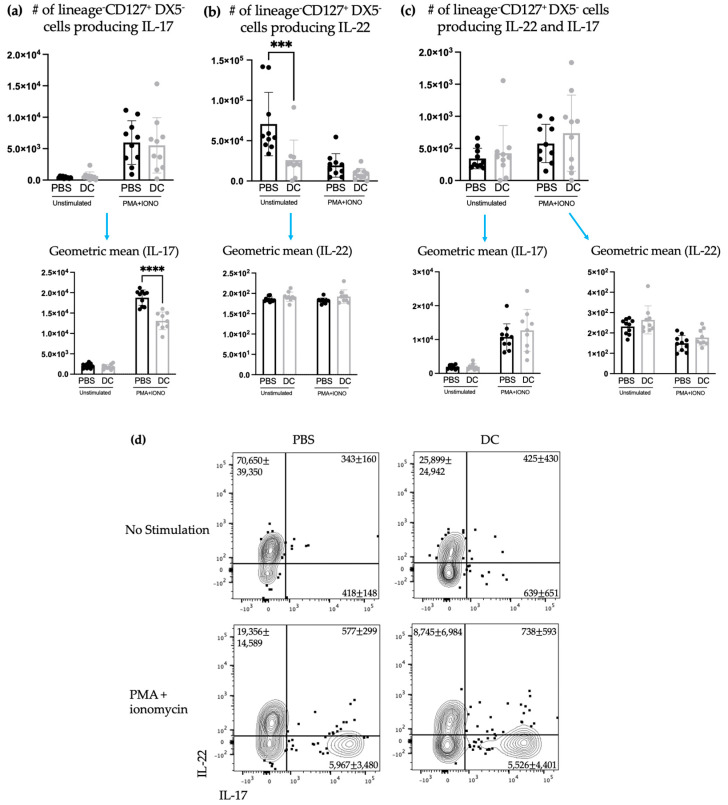
There was no change in the number of ILC3s producing IL-17 and/or IL-22 in the lungs after DC vaccination and challenge with B16F10 melanoma cells. Female C57BL/6 mice (*n* = 10) were inoculated with DC vaccines via hind footpad injection and one week later, they were intravenously administered with 3 × 10^5^ B16F10 cells. Three days later, the lungs were harvested and examined for ILC3-mediated cytokine production. The number of lineage^−^CD127^+^DX5^−^ cells producing (**a**) IL-17, (**b**) IL-22, and (**c**) both IL-17 and IL-22 and their corresponding geometric mean fluorescent intensities was determined using intracellular cytokine staining and flow cytometry. Data were analyzed using a two-way ANOVA test (*p*-values *** < 0.0005). There was no significant difference in the total number of IL-22-producing, IL-17-producing, and IL-22 and IL-17 multi-cytokine-producing lineage^−^CD127^+^DX5^−^ cells. (**d**) Representative dot plots showing the average number of pulmonary CD45^+^lineage^−^CD127^+^DX5^−^ cells producing IL-17 and/or IL-22. The standard deviation and mean are represented by the graph bars and error bars.

**Table 1 ijms-24-02403-t001:** ILC3 subsets defined by flow cytometry analysis.

ILC3 Subset	Flow Cytometry Markers
NCR^+^ ILC3	Lineage cocktail^−^ CD45^+^ DX5^−^ CD127^+^ RORγt^+^ T-bet^+^ NKp46^+^
NCR^−^ ILC3	Lineage cocktail^−^ CD45^+^ DX5^−^ CD127^+^ RORγt^+^ T-bet^−^ NKp46^−^

## Data Availability

Data are available from the corresponding author upon reasonable request.

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
