# Peer review of "The Potential of Dendritic-Cell-Based Vaccines to Modulate Type 3 Innate Lymphoid Cell Populations"

_ijms, 2023, doi:10.3390/ijms24032403_

Round 1

Reviewer 1 Report

Chen et al. explore the impact of adoptive transfer of DC on ILC3 biology. This may be of significant relevance for cancer treatment as ILC3 have been associated with either a pro- or anti-tumour immune response. The mechanisms behind these diverse functional outcomes are still unknown, and addressing these questions is challenging due to the lack of mouse models with a specific lack of ILC3 in the presence of adaptive immunity.

The manuscript from Chen et al. is well written, but even the existing data sets could be used to strengthen the notions raised. 

Specific points:

1) In the appendix all ILC gating strategies for LN, spleen, lungs, intracellular cytokine and intracellular transcription factors (TF) should be shown. Staining controls for lineage markers, CD127 and TF should be added. The CD127 and Lin stain appears quite low. Hence, it is recommended to transform the axes to show more convincing populations.

2)Line 73: In another more recent study it was shown that NKp46+ ILC3 can produce a low amount of IL-17A (Fiancette et al., 2021; PMID:345556884)

3) Clarify in the introduction what DC vaccines are.

4) Double-check on commas (e.g. line 29)

5) Show % of CD45+ cells for Figure 1, 2, 3 and 4.

6) Figure 1b: Show whether PBS vs day 7 is significant or not.

7) Investigate T-bet+ and T-bet+ NKp46- cells in Figure 1, 2, 4 and 6.

8) Analyse gMFI for all cytokine and TF analyses (Figures 3, 5 and 7)

9) The cytokine analysis protocol is not ideal for Figure 7. It would be interesting to know which cytokines are expressed in vivo. In order to do this the authors may harvested the cells and stimulate the cells in vitro with PMA, ionomycin and brefeldin A/monensin for no longer than 3 hours. The issue with the method chosen is that PMA and ionomycin do stimulate cytokine expression on the translational and transcriptional level. The authors added brefeldin A only after one hour in culture and stimulated the cells for a total of 5 hours. This method is acceptable to phenotype cells in the tissues that have the potency to produce the respective cytokines which should be stressed in the text.

10) Discuss this paper in the manuscript: Goc et al., 2021, PMID:34407392

Reviewer 2 Report

In this paper, the authors investigate the interaction between DCs and ILC 3s in a DC vaccine context and the potential role it could play in DC vaccine therapies. They observed an increase in ILC3-derived cytokine production in a tumor free mouse model post DC vaccination. Although changes in cytokine levels were not seen in a tumor model, the ILC3 phenotype and numbers were altered in the lungs suggesting that DC vaccination could have an impact on ILC3s in the tumor challenge model. I found the manuscript well written, with sufficient background and adequate referencing. The materials and results sections were described with detail and easy to follow. The conclusions were well supported by the results and I accept the manuscript in its current form.

Round 2

Reviewer 1 Report

The manuscript of Chen et al. has been much improved. There are only a few minor points the authors should address. 

1. Regarding question 7 the authors have performed analyses on RORgt-negative cells. a) These data in the rebuttal letter should also be added to the manuscript as these ILC1 data nicely support major conclusions of the manuscript. b) I initially intended to suggest an analysis of T-bet- and T-bet+ ILC3, and these data should also be presented and added alike the ILC1 data.

2. Coming back to question 8, I would like to see the gMFI for T-bet added to Figures 3, 5 and 7. This should be for ILC3 and eventually also for ILC1.

3. Across the figure legends the authors should clarify whether NCR is the same as NKp46 (for instance line 157).

4. Line 181: typo: "RO gamma t" 
